# Effects of K_2_TiF_6_ and Electrolyte Temperatures on Energy Consumption and Properties of MAO Coatings on 6063 Aluminum Alloy

**DOI:** 10.3390/ma16051830

**Published:** 2023-02-23

**Authors:** Xiaomeng Xie, Erhui Yang, Ziying Zhang, Wu Wei, Dong Li, Xiaolian Zhao, Ruixia Yang, Weizhou Li

**Affiliations:** 1School of Resources Environment and Materials, Guangxi University, Nanning 530004, China; 2School of Materials Science and Engineering, Xiamen University of Technology, Xiamen 361024, China; 3School of Mechanical and Marine Engineering, Beibu Gulf University, Qinzhou 535011, China

**Keywords:** micro-arc oxidation, dipotassium titanium hexafluoride, electrolyte temperature, energy-consumption, corrosion resistance

## Abstract

To decrease energy consumption and improve the performance of micro-arc oxidation (MAO) films on 6063 Al alloy, a policy of K_2_TiF_6_ additive and electrolyte temperature control was adapted. The specific energy consumption relied on the K_2_TiF_6_ additive and more particularly on the electrolyte temperatures. Scanning electron microscopy demonstrates that electrolytes with 5 g/L K_2_TiF_6_ can effectively seal the surface pores and increase the thickness of the compact inner layer. Spectral analysis shows that the surface oxide coating consists of γ-Al_2_O_3_ phase. Following 336 h of the total immersion process, the impedance modulus of the oxidation film, prepared at 25 °C (Ti5-25), remained 1.08 × 10^6^ Ω·cm^2^. Moreover, Ti5-25 has the best performance/energy-consumption ratio with a compact inner layer (2.5 ± 0.3 μm). This research found that the time of the big arc stage increased with the temperature, resulting in producing more internal defects in the film. In this work, we employ a dual-track strategy of additive and temperature providing an avenue to reduce the energy consumption of MAO on alloys.

## 1. Introduction

As a non-ferrous structural material, aluminum alloy is one of the most widely used in industry, for instance, in electronic communication, textiles, automobile, shipping, aviation and other fields. Nevertheless, weak corrosion and wear resistance of aluminum alloy have always restricted its development. Micro-arc oxidation is a surface processing technique with great promise. The ceramic oxide film formed in situ on aluminum alloy by using environmentally friendly electrolyte [1], improves the properties of aluminum alloy [2,3,4,5]. The MAO is carried out under the condition of high current and high voltage, so this technology has the disadvantages of high energy consumption and low energy utilization efficiency, which hinders its further development and application [6]; At the same time, owing to the use of traditional MAO basic electrolytes, there are a lot of pores and cracks in these MAO films [7,8,9], allowing the corrosion medium to easily penetrate into the substrate, which causes corrosion. Therefore, finding a method that can not only reduces energy consumption, but also improves corrosion resistance has become a requirement in the development of MAO technology.

At present, MAO often balances performance and energy consumption by adjusting power parameters and optimizing the electrolyte formula. Changing the current output mode, the current step-down mode uses about 10% less energy than the constant current mode [10]. Compared with the MAO process, the low-energy MAO process with relatively low voltage (~100 V) can save about 57.0% of energy [11]. Changing the ratio of the anode current to a cathode current (Rpn), the transition to soft plasma [12], changing the power supply mode [13], decreasing the duty cycle and oxidation time [14] and preparing an anodic oxidation prefabricated film [15,16], all of these can cut down the energy consumption of the MAO treatment. Nevertheless, the gain in the energy utilization rate is not obvious because of the regulating electrical parameters. In contrast, optimizing the electrolyte composition can significantly reduce the energy consumption and improve the film’s properties. Cheng et al. [17] found that the energy consumption of the MAO process on Al-Cu-Li alloy in electrolytes containing 5, 32 and 56 g·L^−1^ NaAlO_2_ were 13.1~16.7, 3.3~4.8 and 1.1~1.5 kW·h·m^−2^·μm^−1^, respectively. Following the investigation, we have not found anyone who has studied the influence of K_2_TiF_6_ additive on the energy consumption of micro-arc oxidation. Most MAO coatings prepared on aluminum alloys are mainly composed of alumina and other solution-related compounds, such as SiO_2_, aluminum phosphate and mullite [14,18]. These compounds do not provide enough corrosion protection for the substrate, which make the substrate corrode easily [3,7]. Rahmati et al. [19] studied the coating formation, pore sealing mechanism and properties of AZ31 magnesium alloy MAO in silicate electrolytes containing K_2_TiF_6_. Adding K_2_TiF_6_ increased the average thickness of the MAO film, reduced the porosity and formed a thin fluoride-rich passive layer. During the discharge process, TiO_2_ particles entered the discharge channel to form a thick and dense coating. The incorporation of SiO_3_^2−^ formed amorphous silica in the MAO film, which also sealed some pores in the coating. Therefore, the coating still had high long-term corrosion resistance. Therefore, the addition of K_2_TiF_6_ could form a denser and thicker MAO film on aluminum alloy, thus improving the corrosion resistance of aluminum alloy and reducing the energy consumption. At the same time, the film contained titanium dioxide, which gave it greater application prospects for biologically related materials [20], as well as photocatalysis [21] and anticorrosion properties [22].

During the MAO process, the electrolyte temperature also influenced the properties of the MAO film. The MAO electrolyte temperature significantly affected the growth and thickness of the ceramic coating [23]. In low-temperature alkaline aluminate electrolytes, more α-Al_2_O_3_ was formed in the coating, resulting in a high wear resistance and low porosity [24]. Mohannad et al. [25] researched the micro-morphology of MAO film on 6061 aluminum alloy in an alkali silicate solution at different temperatures. It was indicated that the electrolyte temperature affected the characterization of the coating surface. In the electrolyte with a low temperature, a volcano-like structure made up of accumulated particles was formed on the coating surface. The MAO alumina ceramics prepared in the electrolyte with a high temperature were rough, thin and contained grainy spherical hollow bulgy structures, while the pore density, silicon content and infrared emissivity were all high. Therefore, the electrolyte temperature was markedly affected by the MAO coating.

Sreekanth [26,27] studied the effect of adding K_2_TiF_6_ to silicate electrolytes on the morphology and corrosion resistance of ZM21 and AZ31 magnesium alloys, but did not pay attention to the electrolyte temperature. Tang [28,29] researched the effect of adding K_2_TiF_6_ on the morphology and corrosion resistance of MAO coating on 2A70 aluminum alloy and AZ91D magnesium alloy at the electrolyte temperature of 40 °C, but did not consider the impact of the electrolyte temperature. Fernandez [30] recently discussed the influence of K_2_TiF_6_ on the morphology, wear resistance and corrosion resistance of the MAO coating of a secondary casting Al-Si alloy when the electrolyte temperature was controlled at 7 °C. Unfortunately, the influence of the electrolyte temperature was not investigated in depth.

This paper aims to investigate the morphology, structure, energy consumption and properties of MAO film on 6063 Al alloy produced by adding 5 g/L K_2_TiF_6_ and controlling the average electrolyte temperature at 15 °C, 25 °C and 35 °C, respectively. At the same time, the synergistic effect of the electrolyte temperature and K_2_TiF_6_ on the growth process and mechanism of the coating is also discussed. In this study, combining the temperature with the electrolyte formula to obtain a high-performance MAO film with a low energy consumption on aluminum alloy laid the foundation for the early application of this process in industry.

## 2. Experiment

### 2.1. Experimental Materials and Film Fabrication

MAO films were prepared on oblong blocks (20 × 15 × 3 mm^3^) of 6063 Al alloy (Si 0.20%, Cu 0.015%, Fe 0.25% and Al balance). All specimens were successively polished with 400#, 1000#, 1500# and 2000# sandpaper, until they acquired a surface roughness of ≈0.1 µm. Then, the specimens were ultrasonically washed in 99.9% anhydrous ethanol followed by warm blowing drying. The dilute alkaline solution was composed of: Na_2_SiO_3_ (10 g·L), Na_3_PO_4_ (10 g·L), NaF (2 g·L) and K_2_TiF_6_ (0 or 5 g·L) as an additive. The average temperatures of the electrolyte were: 15 °C, 25 °C and 35 °C. The composition, temperature, conductivity and pH of the different electrolytes are reported in Table 1. The pretreated aluminum alloy specimens were oxidized for 10 min at a constant current density of 10 A·dm^−2^, a 30% duty cycle and a frequency of 400 Hz. Then, the MAO samples were ultrasonically washed in de-ionized water followed by warm blowing drying.

### 2.2. Coating Characterizations

During the MAO treatments, the discharge behaviors were recorded using a commercial digital camera (Handycam FDR-AX100), operating at 100 frames per second. The thickness of the MAO films was the presented average of 10 measurements taken at diverse positions and measured with a TT260B coating eddy current meter. Meanwhile, the phase composition of the films was confirmed by employing a MiniFlex600-C X-ray diffractor (XRD) at a speed of 6°/min, at 40 kV and 150 mA, in the 2θ range of 20° to 80°. We analyzed the morphologies, cross-sections and element composition on the films with a scanning electron microscope (SEM/EDS, SU-8020/X-Max 80). The coating Ti5-15 surface was analyzed using X-ray photoelectron spectroscopy (XPS, Thermo ESCALAB 250XI), and then the XPS curve was fitted using XPSPEAK4.1 software, Version 4.1 November 2000, Raymond Kwok, Shatin, Hong Kong, China.

### 2.3. Energy Consumption

Through the power signal acquisition system, the specific data of voltage and current changes with the reaction time were obtained, and then the unit energy consumption was calculated according to Equation (1) [31], taking the average of three parallel samples.
(1)$$\rho =\frac{\int_{0}^{T}V_{t}I_{t}dt}{SL}.$$
where ρ is the energy consumption to generate a unit volume of the micro-arc oxide film layer (kw·h·m^−2^·μm^−1^), *V_t_* as transient voltage (V), *I_t_* as *Transient* current (A) and *T* as duration time in the MAO process, S as the specimen superficial area (m^2^) and *L* as film thickness (μm).

### 2.4. Corrosion Behavior of the Films

In 3.5 wt% NaCl solution, we used an electrochemical workstation (CS350, CorrTest) to measure the potentiodynamic polarization (PDP) with a 10 mV/s scanning rate to investigate the corrosion performance. The tests used a saturated calomel electrode and platinum plate as the reference electrode and counter electrode, and 1 cm^2^ of film that was exposed in the NaCl solution as the working electrode. In addition, we used electrochemical impedance spectroscopy (EIS) to investigate the electrochemical behavior of the MAO films. The disturbance amplitude was 10 mV and the frequency ranged from 10^5^ Hz to 0.01 Hz. Some EIS tests were carried out after 336 h of immersion to research the long-term corrosion properties of the specimens. The Zview software was used to analyze the test data.

## 3. Results

### 3.1. Evolution of Voltage and Discharge

According to the voltage response curves and the sparks in the reaction stage, we distinguished four typical discharge stages in the MAO treatment: stage I (anodic oxidation), stage II (spark discharge), stage III (micro-arc) and stage IV (big arc) [32]. During stage I, quantities of gas were generated, a barrier film was formed and the metallic luster gradually vanished from the specimens’ surface. In stage II, the voltage rose linearly and rapidly, which was maintained for a short time. A large number of fast-walking small white sparks were discovered and the voltage increase rate declined. Entering stage III, these sparks turned yellow, augmented in size and reduced in quantity. Then, the voltage achieved a steadier level and sustained a slow rise. During stage IV, some large arcs with a longer lifetime were observed in the local regions of the specimens.

Figure 1 indicates the evolution of voltage and the spark discharge features of samples Ti0-15, Ti5-15, Ti5-25 and Ti5-35. The effect of the diverse electrolytes to the voltage response curves is prominent [29]. From Figure 1a, samples Ti5-15, Ti5-25 and Ti5-35 with the addition of 5 g/L K_2_TiF_6_ were different from sample Ti0-15 without K_2_TiF_6_. In stage I, the voltage of Ti0-15 increased faster than that of Ti5-15, Ti5-25 and Ti5-35, while the voltage of Ti5-15 increased the slowest. Nevertheless, Ti5-15, Ti5-25 and Ti5-35 demonstrated a higher working voltage than Ti0-15 in stage III. The termination voltages of Ti0-15, Ti5-15, Ti5-25 and Ti5-35 were about 441 V, 475 V, 478 V and 450 V, respectively.

From Figure 1b, sparks appeared on Ti5-15, Ti5-25 and Ti5-35 surface within 90 s (~322 V, ~296 V and ~281 V), while sparks appeared on the Ti0-15 surface at 111 s (~366 V), respectively. These consequences exhibited that a dielectric breakdown had occurred on the specimens with 5 g/L K_2_TiF_6_, as opposed to without K_2_TiF_6_. The metal surface property, the component and conductivity of the solution strongly influenced the breakdown voltages [28,33]. The discharge phenomenon of Ti0-15 and Ti5-15 was chiefly composed in spark discharge stage and the micro-arc stage that commenced after 130 s. When the average electrolyte temperature was 25 °C, the big arc stage started after 480 s. The big arc stage initially began after 390 s when the electrolyte temperature was 35 °C. The start time of the big arc stage advanced with the increase of the electrolyte temperature, causing a longer big arc stage reaction time.

### 3.2. Microstructure of the Films

It can be observed from Figure 2a that the micropores on the Ti0-15 surface are relatively smaller and fewer because of the small dimension of the discharges. As can be observed in Figure 2, films Ti5-15, Ti5-25 and Ti5-35, with the addition of 5 g/L K_2_TiF_6_ had larger dimension pores than film Ti0-15 without K_2_TiF_6_, that should be ascribed to the involvement of TiO_2_ particles in the electrochemical reaction, which caused the coating thickness and working voltages to increase (Figure 1a). The coating surface of Ti5-15 was compact and uniform and the large pores were full of oxide. From Figure 2c,d, there are obvious porous regions, where big pores were distributed in concentration and appeared on the surface when the electrolyte temperatures rose to 25 °C. These porous areas might be formed by the generation of constant and big dimension arcs in defects in the coating occurring in stage IV. The operation time of stage IV was lengthened with the higher electrolyte temperatures, which brought about the increase of the coverage area of the porous regions and more rough coating surfaces. In addition, the quantity and size of the sparks in the non-porous regions were reduced, because more electric energy was concentrated in the porous regions.

The EDS results of surfaces of films Ti0-15, Ti5-15, Ti5-25 and Ti5-35 are shown in Table 2. Film Ti0-15 without K_2_TiF_6_ was mostly made up Al and O. The element composition of films Ti5-15, Ti5-25 and Ti5-35 with the addition of 5 g/L K_2_TiF_6_ all mainly contained Al, O and Ti. With the addition of 5 g/L K_2_TiF_6_, the values of the Al mass percent decreased on the surfaces, while that of Ti increased. The values of the Ti mass percent decreased on Ti5-15, Ti5-25 and Ti5-35 surfaces with the elevation of the electrolyte temperature indicating a lower incorporation of Ti in the films [29]. However, the incorporation of the compositions from the solution increased with the elevation of the electrolyte temperature. The lower viscosity with the elevation of the electrolyte temperature led to it becoming easier for the hotter electrolyte to flow to the reaction surface and augmented the direct electrolyte-surface interaction [25]. Moreover, Ti content in Ti5-15, Ti5-25 and Ti5-35 was higher than that in the MAO coatings prepared by Tang [28] and Arunnellaiappan [33] on 2A70 and AA7075 aluminum alloys, by adding potassium fluotitanate. This showed that TiO_2_ sol particles could easily enter the coating by this method.

A cooling device was used to keep the cooling water below 15 °C, and 5 g/L K_2_TiF_6_ was added for the MAO treatment for 120 and 180 s to study the incorporation mechanism of the solution’s components. The surface micromorphology of these two coatings were shown in Figure 3. Following treatment for 120 s, it was in the spark discharge stage (the second stage) that the detected O and F content on the surface exceeded that of the other electrolyte elements (Table 3), that is, other the elements extracted from the electrolytes were very low, indicating generated aluminum fluoride and alumina. In other words, a thin fluorine barrier layer was generated early in the MAO [19]. Following treatment for 180 s and just entering the micro-arc oxidation stage (the third stage), the F content dropped, and the content of other elements increased with the increase of thickness (Table 3), Forming more and larger discharge channel anions. Therefore, a large number of elements in the electrolyte were pulled into the channels of the outer coating after 180 s.

Figure 4 displays the cross-sectional morphology and the EDS line scan results of different MAO films. From Figure 4a, there are cavities of different dimensions as a band at the substrate/film interface of Ti0-15. The connected large cavities are formed by the strong spark discharge at the substrate/film interface [5]. There are also plenty of pores of different dimensions and cracks in Ti0-15. The micro discharges and gas bubbles resulted in the formation of micropores. The thermal stress that was formed when the molten oxide solidified rapidly and brought about the generation of microcracks [19,20]. Ti5-15 with a fine continuity, fewer defects and thicker compact inner layer was produced at 15 °C by adding 5 g/L K_2_TiF_6_ (Figure 4b). The micro discharges of stage III were weak and well-proportioned, which could repair the defects and fill the pores in the films quickly, so the denser films could be acquired by lengthening the stage III operation time. In Figure 4c,d, the quantity and size of the pores of Ti5-25 and Ti5-35 were augmented because of the extension of the stage IV operation time with the increase in electrolyte temperature. The production of big pores and cracks occurred due to the formation of long-lasting, uneven and strong discharge breakdown in the films. The greater thermal stresses were concentrated near the big discharge sparks, so it was hard to repair the defects and fill the pores. This further showed that the longer operation time of stage III is helpful for forming a dense film, while the longer action of stage IV produces more internal defects in the film.

The distribution of the elements of the cross-sectional Ti0-15, Ti5-15, Ti5-25 and Ti5-35 obtained by the EDS line scan are shown in Figure 4a’–d´. Al, O and Ti were dispersed nearly evenly across the thickness of the films. The incorporation of the F element obviously increased at the substrate/film interface for all films. The F^−^ ions were easily congregated at the substrate/film interface and generated the AlF_3_ with reaction of Al3^+^ and F^−^ ions [19]. The mass ratio of aluminum and oxygen in the coating were reduced, while the mass ratio of aluminum in Ti augmented with the addition of 5 g/L K_2_TiF_6_. When the electrolyte temperature increased, the mass fraction of Al, Ti and O in Ti5-15, Ti5-25 and Ti5-35 coatings decreased. Consistent with the results of the surface analysis.

The thickness of the films significantly increased with 5 g/L K_2_TiF_6_ additive and the elevation of the electrolyte temperature due to the increased voltage that accelerated the development of the films. As evidently recognizable from the cross-section micrographs (Figure 4a–d), the inner and outer layers made up the total MAO film. The compact inner layers of films Ti5-15, Ti5-25 and Ti5-35, with the addition of 5 g/L K_2_TiF_6_ were thicker than film Ti0-15 without K_2_TiF_6_. The coating thicknesses of films Ti0-15, Ti5-15, Ti5-25 and Ti5-35 were 5.8 ± 0.7 µm, 12.3 ± 1.0 µm, 21.3 ± 1.5 µm and 47.9 ± 4.1 µm by TT260B coating eddy current meter, however, the thickness of the compact inner layer of Ti5-15 was the largest, which was 3.9 ± 0.3 µm according to the cross-sectional SEM analysis, as shown in Table 4. The thickness increment of the MAO film obtained by combining the temperature and electrolyte formula is higher than that of the coating prepared by adding K_2_TiF_6_ only [19,29]. This result showed that the growth rate of the coating prepared using this method is greatly improved. Figure 5 revealed the unit energy consumption of Ti0-15, Ti5-15, Ti5-25 and Ti5-35. The energy consumption of Ti5-15, Ti5-25 and Ti5-35 was significantly lower compared to Ti0-15. In Figure 5, the unit energy consumption of Ti5-35 (1.71 kw·h·m^−2^·µm^−1^, only 13% of the energy consumption of Ti0-15) was lower. Compared with most of the reported research energy consumption, the energy consumption reduction effect is achieved [10,11,12,13,14,15,16]. It demonstrated that both 5 g/L K_2_TiF_6_ additive and the elevation of electrolyte temperature could increase the coating thickness to reduce the energy consumption.

### 3.3. Phase Composition

Figure 6 showed the XRD patterns in which films Ti0-15, Ti5-15, Ti5-25 and Ti5-35 were displayed. It can be observed that the major phases of Ti0-15, Ti5-15, Ti5-25 and Ti5-35 were γ-Al_2_O_3_, the amorphous phase and Al because of the penetration of XRD through the film to the substrate [25]. With 5 g·L^−1^ K_2_TiF_6_ additive and the increase in the electrolyte temperature, the descent of the XRD peak intensity of Al could be caused by the increase of the coating density and thickness [34]. Nevertheless, the peak intensity of γ-Al_2_O_3_ decreased and the diffuse scattered peak of the amorphous phase at 2θ within 10–35° increased with 5 g·L^−1^ K_2_TiF_6_ additive and the increase in the electrolyte temperature. In other words, Ti0-15 was composed of higher γ-Al_2_O_3_ but was lower than Ti5-15, Ti5-25 and Ti5-35 in the amorphous phase content. The element Ti was detected in Ti5-15, Ti5-25 and Ti5-35 by EDS, but there were no crystal phase peaks relevant to titanium oxides. This could be attributed to their amorphous structure or low content or the combination of Ti and γ-Al_2_O_3_ [19,29]. As shown in Figure 7, the XPS spectrum of Ti 2p had a Ti 2p_3/2_ peak at 458.7 eV and a Ti 2p_1/2_ peak at 464.3 eV, which was consistent with the binding energy of TiO_2_ [33]. It was further proved that the MAO coating prepared by adding K_2_TiF_6_ contains TiO_2_.

### 3.4. Corrosion Resistance of the Films

The fitting parameter value and PDP of films Ti0-15, Ti5-15, Ti5-25 and Ti5-35 are displayed in Figure 8 and Table 5. From fitting dates and potentiodynamic polarization curves, the self-corrosion current density (icorr) of the MAO coating with added potassium fluotitanate dropped, and the self-corrosion potential (Ecorr) rose. The self-corrosion current density and corrosion rate of Ti5-15 film were both minimal, which were 7.238 × 10^−5^ mA/cm^2^ and 0.071 mm/a, respectively. Moreover, Ecorr of Ti5-15 coating achieved −1.183 V, indicating that Ti5-15 film has a good corrosion resistance. 

Bode plots of films Ti0-15, Ti5-15, Ti5-25 and Ti5-35 are displayed in Figure 9a–d. Figure 9e shows the variation of impedance values |Z|_0.01Hz_ at 0.01 Hz with prolonging of the soaking time. All impedance values reduced after a 24 h immersion. It should be noted that Ti0-15 descended quickly, and the impedance value |Z|_0.01Hz_ dropped two orders of magnitude after a 336 h immersion, demonstrating exacerbation of the film [3]. On the contrary, |Z|_0.01Hz_ of Ti5-15, Ti5-25 and Ti5-35 slightly increased with the prolonging of the immersion time in the soaking period of 24 h. Maybe this is because of the small dimension and shallow depth of the pores on the surface of Ti5-15, Ti5-25 and Ti5-35, which could be occluded directly by the corrosion products [35]. For this reason, the penetration of corrosion solution into the films was hindered. Following 336 h of the total immersion process, the impedance values |Z|_0.01Hz_ of Ti5-15 and Ti5-25 remained above 10^6^ Ω·cm^2^, demonstrating that Ti5-15 and Ti5-25 displayed good corrosion protection. In addition, the low-frequency impedance values of Ti5-15, Ti5-25 and Ti5-35 decreased more slowly than those of the micro-arc oxidation coatings prepared by Rahmati [19] and Fernandez [30] on AZ31 Mg alloy and secondary cast Al-Si alloy. It can be considered that the long-term corrosion resistance of Ti5-15, Ti5-25 and Ti5-35 coatings is better.

The Nyquist diagram (Figure 9g–g’’) and phase (Figure 9f) diagrams of films Ti0-15, Ti5-15, Ti5-25 and Ti5-35 after a 336 h immersion were fitted using three kinds of electrical equivalent circuits (Figure 10) by Zview software. In addition, Table 6 reports the data of the corresponding fitting electrochemical parameters. R_S_ is the uncompensated solution resistance, R_O_, R_I_ and R_ct_ are the resistance of the outer layer, inner layer and charge transfer, respectively. CPE_O_, CPE_I_ and CPE_ct_ are the capacitance of the outer layer, inner layer and charge transfer, respectively. Y_O_, Y_I_ and Y_ct_ are the admittance constant n_O_, n_I_ and n_ct_ (0 < *n* < 1) are the index of CPE_O_, CPE_I_ and CPE_ct_. Furthermore, inductor L and R_L_ were used to indicate the pitting corrosion in the films.

Following 336 h of immersion, the inductive loop displayed at the low-frequency region for Ti0-15 (Figure 10c). This showed that the beginning of the corrosion of aggressive solution at the substrate and the MAO film should not protect the Al alloy substrate any more. The Nyquist diagram (Figure 9g’), phase diagram (Figure 9f) and equivalent circuit (Figure 10b) diagram of Ti5-35 indicated three-time constants. Thus, it could be deduced that the penetration of the aggressive solution into the film/substrate interface through the micro-pores, indicated the onset of the interface corrosion reaction. Two-time constants were indicated by the electrical equivalent circuit plots of Ti5-15 and Ti5-25 in Figure 10a. This verified the bi-layer structure of the MAO films and the better anti-corrosion property of Ti5-15 and Ti5-25 [7,19]. It can be seen in Table 4, that the resistance values R_O_ at high frequencies of Ti5-15, Ti5-25 and Ti5-35 were 1.01 × 10^6^, 1.3 × 10^5^ and 319 Ω·cm^2^, respectively, illustrating the resistance value of the outer porous layer was dependent on the composition and microstructure. However, the resistance value R_I_ at low frequencies of Ti5-15 and Ti5-25 remained high, which were 5.94 × 10^6^ and 9.52 × 10^5^ Ω·cm^2^, respectively, which was far higher than R_O_. It could be implied that the more compact inner layers were the decisive factor of the anti-corrosion performance of the whole film. 

As shown in Figure 11a, after soaking in 3.5% sodium chloride solution for 336 h, Ti0-15 surface had a number of defects and corrosion spots, indicating the dissolution and breakdown of the film initiated in a corrosion pit during the immersion. It was not possible to observe any corrosive defects on the coating surfaces because of the color of the Ti5-15, Ti5-25 and Ti5-35 surfaces were black (Figure 11b,d). The surface micro-graphs of Ti0-15, Ti5-15, Ti5-25 and Ti5-35 were gained by SEM (Figure 11e–i), so that we could observe the corrosion phenomenon clearly. Following immersion for 336 h, Ti0-15 was seriously corroded (Figure 11e). From Figure 11f–i, a black defect was observed on the Ti5-35 surface, while Ti5-15 and Ti5-25 surfaces had almost no obvious abnormalities. This demonstrated the better anti-corrosion property of Ti5-15 and Ti5-25. A lot of white corrosion products were visible on Ti5-15 and Ti5-25, while a few were observed on the Ti5-35 surface, which was consistent with the impedance values of the analysis results.

### 3.5. Influence of the Electrolyte Temperature on the Coating Growth

Hussein et al. [36] indicated that the MAO discharges could be divided into three types, and Cheng et al. [5] demonstrated that should be four. Discharge type B happens on the alloy/film interface and is strong, discharge type A happens on the film/solution interface, discharge type C happens in the film’s surface defects and discharge type D happens in the inner closed pores of the film, respectively, the latter three discharge types belong to the weak discharge. According to the above study and discussions, the MAO film growth mechanism is shown in Figure 12. During stage I (Figure 12a–d), an anodic oxide film was formed by the reaction of Al^3+^ from the alloy and O^2−^ from the solution on the surface and gas was produced [31,37]. F^−^ migrated from the solution/film interface to the film/alloy interface faster than O^2−^ and hydroxide ions [38], and they generated a compact barrier layer on the film/alloy interface [19]. Following the addition of 5 g·L^−1^ K_2_TiF_6_ to the solution, the concentration of F increased. Moreover, the migration rate of the anion augmented with the increase of the electrolyte temperature. So that, the anodic oxide film was formed preferentially by adding 5 g·L^−1^ K_2_TiF_6_ and the increase of the electrolyte temperature. This could be proved by the fact that the metallic luster of the Ti-35 surface gradually disappeared after 15 s at the earliest (Figure 1b). Then, the breakdown first occurred on the surface of Ti-35 with a thicker anodic oxidation film (Figure 12h).

As shown in Figure 12i–l, the micro-arc stage (stage 3) was the main growth period of the MAO film layer. Within 6 min, the amount and dimension of the discharge channels augmented, type B discharge played a major role, and there were a few types A and C discharges. Once the coating thickened, the number of discharge decreased, but it was more intensive, resulting in large pores on the surface [32]. The coating growth rate increased with the 5 g·L^−1^ K_2_TiF_6_ additive and an increase of electrolyte temperature. Titania colloidal particles were formed by hexafluorotitanate hydrolysis [28]. When the discharge started, these TiO_2_ colloidal particles could be drawn into the discharge channels and absorbed and accumulated [29]. The participation of TiO_2_ colloidal particles in the reaction increased the coating growth rate [33]. 

Discharge phenomena of Ti0-15 and Ti5-15 were chiefly composed of the spark discharge and the micro-arc. However, the formation of cavities in Ti0-15 without K_2_TiF_6_ provided a condition for discharge type D, leading to holes and an increase in cavities (Figure 12m,q). At the temperature of 15 °C, the discharge sparks were distributed evenly on Ti5-15 surface (Figure 12n). The liberated heat of Ti5-15 was dissipated uniformly from the coating to the solution. It illustrated that the lower temperature of the electrolyte helped to form a dense and thick film (Figure 12r). The discharge of Ti5-25 entered into the big arc stage at 480 s when the electrolyte temperature rose to 25 °C. Therefore, the time of Ti5-25 entering the large arc discharge was short, leading to less aggregation or merging of the arc discharge, which decreased the defects in the film (Figure 12o,s). The starting time of the big arc stage of Ti5-35 was earlier at about 390 s, when the electrolyte temperature rose to 35 °C. Congregation and merging of the arc discharge was obvious by prolonging the big arc stage (Figure 12p). The big discharge sparks were more intensely unevenly distributed, resulting in the concentration of the liberated heat around Ti5-35. For this reason, the electrolyte near Ti5-35 was heated. At this higher temperature, the TiO_2_ colloidal particles in the electrolyte tended to aggregate and became unstable [39]. Thus, the compactness of the film was reduced and produced more internal defects (Figure 12t). It could illustrate that the start time of stage IV was earlier with the increase of the electrolyte temperature, resulting in the longer reaction time of stage IV.

## 4. Conclusions

The titanium dioxide colloidal particles could be driven into the discharge channels by an electric field in a silicate-phosphate mixed based solution containing 5 g/L K_2_TiF_6_. Thus the addition of 5 g/L K_2_TiF_6_ can seal some large holes in the film and thicken the film. The occurring time of the big arc stage advanced with the increase of the electrolyte temperature, causing a longer reaction time of the big arc stage, resulting in more rough coating surfaces and more internal defects. The thickness of the entire coating of Ti5-35 was larger than 47.9 ± 4.1 μm. Therefore, the unit energy consumption of Ti5-35 was lower than 1.71 kw·h/(m^2^·µm), which was only 13% of the unit energy consumption of Ti0-15. Both 5 g·L^−1^ K_2_TiF_6_ additive and the elevation of the electrolyte temperature could increase the coating thickness and improve the energy utilization.

The resistance value R_I_ at low frequencies of Ti-15, Ti-25 and Ti-35 were all higher than the resistance value R_O_ at high frequencies, indicating that the more compact inner layers were the decisive factor in the anti-corrosion performance of the whole film. Following soaking for 336 h in 3.5% sodium chloride corrosive medium, a black corrosion defect was observed on the Ti5-35 surface, while Ti5-15 and Ti5-25 surfaces had almost no obvious abnormalities. Thus Ti5-25 prepared at 25 °C with the addition of 5 g/L K_2_TiF_6_ has the best performance/energy-consumption ratio. 

The lower electrolyte temperature contributed to prolonging the micro-arc oxidation stage reaction, forming a coating with a fine continuity, fewer defects and a thicker compact inner layer. The operation time of stage IV was lengthened with the higher electrolyte temperatures, which led to more obvious phenomenon of aggregation or merging of the arc discharge and the increase of the coverage area of the porous regions and more rough coating surfaces.

## Figures and Tables

**Figure 1 materials-16-01830-f001:**
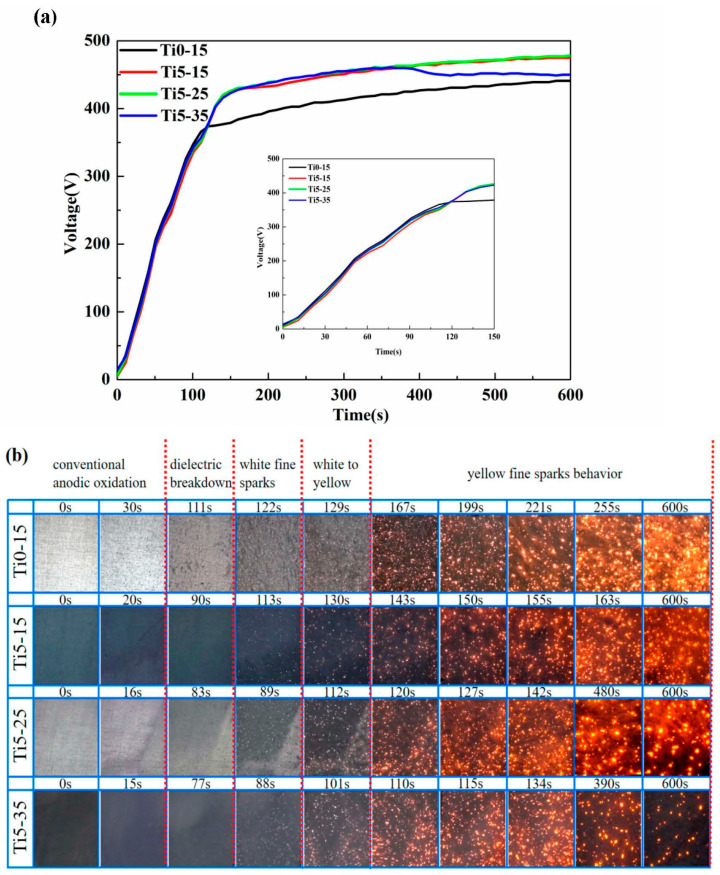
(**a**) Voltage-time curves and (**b**) evolution of the micro-discharge of Ti0-15, Ti5-15, Ti5-25 and Ti5-35 micro-arc oxidation for 10 min.

**Figure 2 materials-16-01830-f002:**
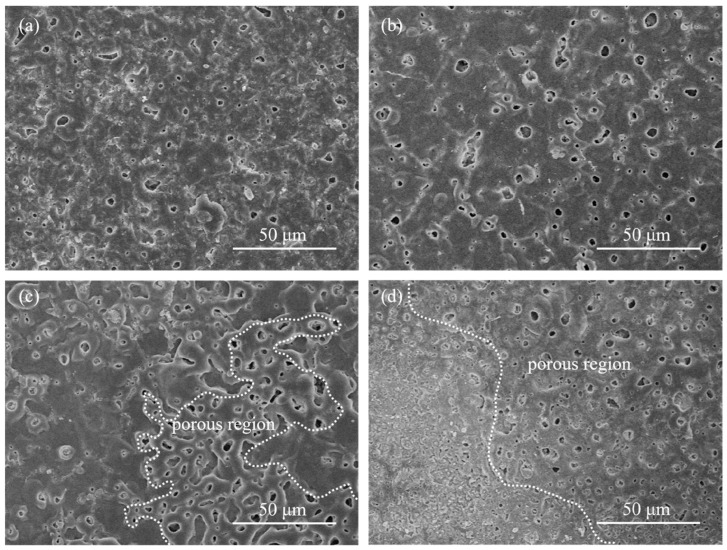
Surface morphology of the MAO film layers of different micro-arc oxidized samples for 10 min: (**a**) Ti0-15, (**b**) Ti5-15, (**c**) Ti5-25 and (**d**) Ti5-35.

**Figure 3 materials-16-01830-f003:**
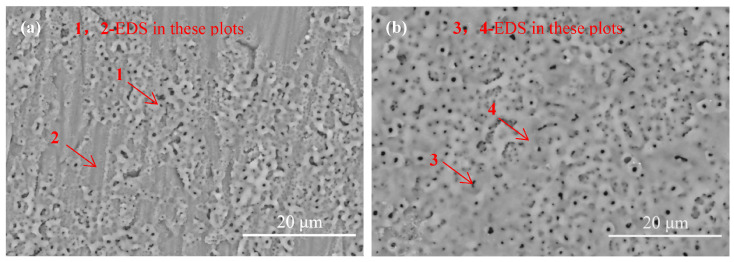
Surface micromorphology of the samples with different treatment times: (**a**) 120 s and (**b**) 180 s.

**Figure 4 materials-16-01830-f004:**
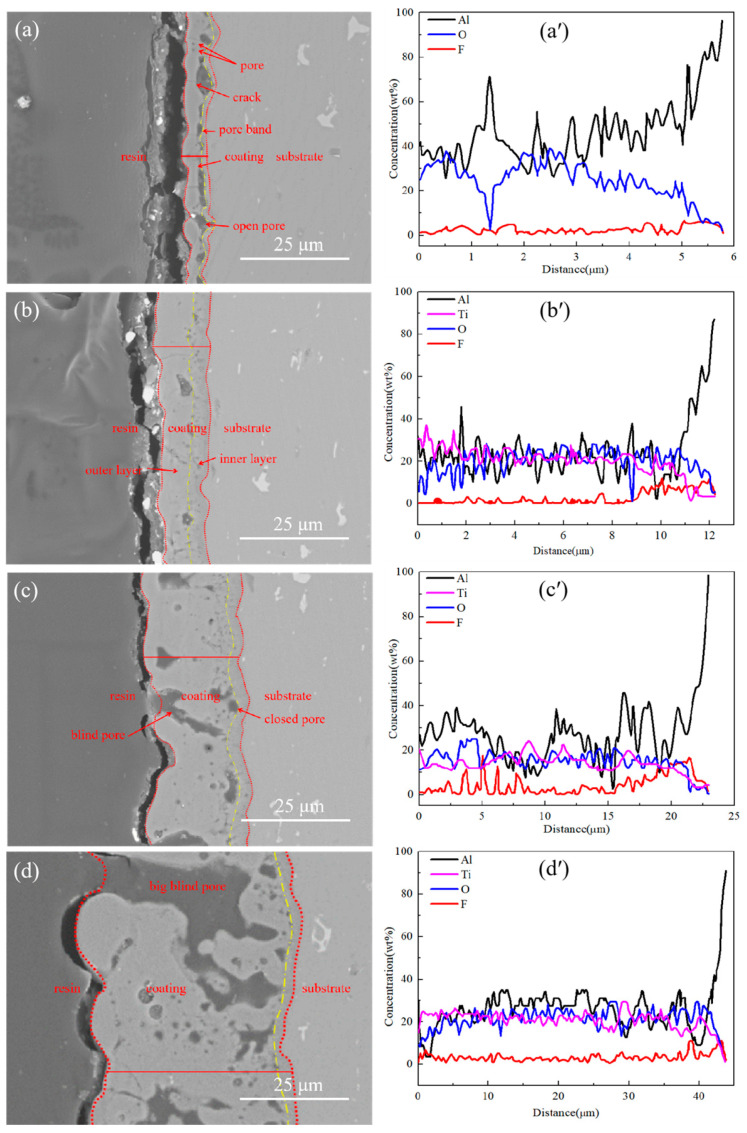
Cross-sectional morphology of MAO films after micro-arc oxidation for 10 min and the elements line distribution examined by EDS: (**a**,**a**’) Ti0-15, (**b**,**b**’) Ti5-15, (**c**,**c**’) Ti5-25 and (**d**,**d**’) Ti5-35.

**Figure 5 materials-16-01830-f005:**
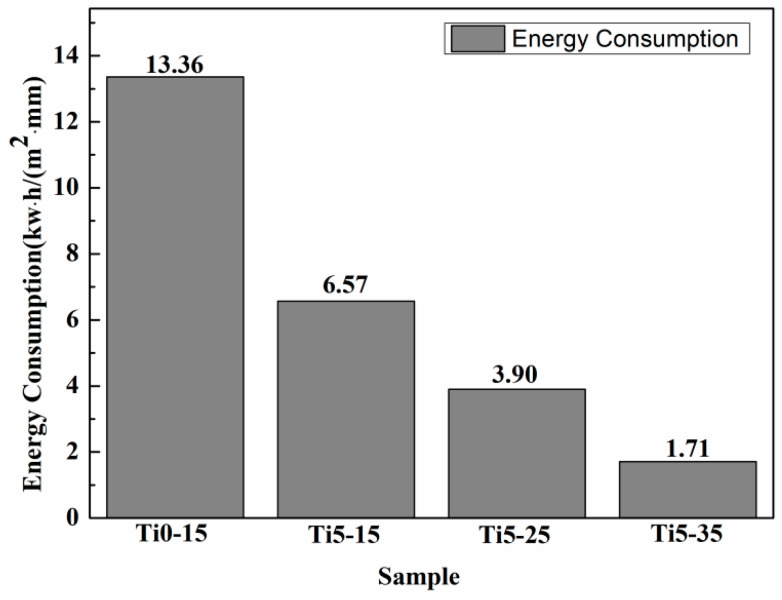
Energy consumption (ρ) for 10 min of micro-arc oxidation for different samples.

**Figure 6 materials-16-01830-f006:**
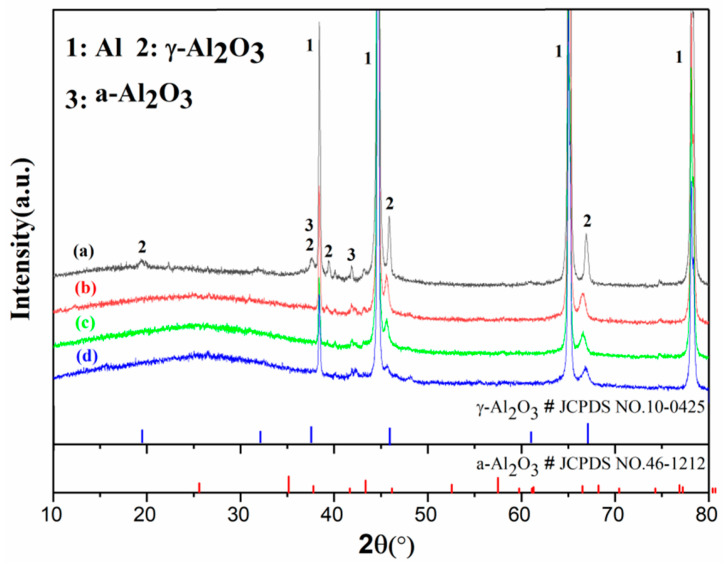
XRD pattern of different MAO films: (**a**) Ti0-15, (**b**) Ti5-15, (**c**) Ti5-25 and (**d**) Ti5-35.

**Figure 7 materials-16-01830-f007:**
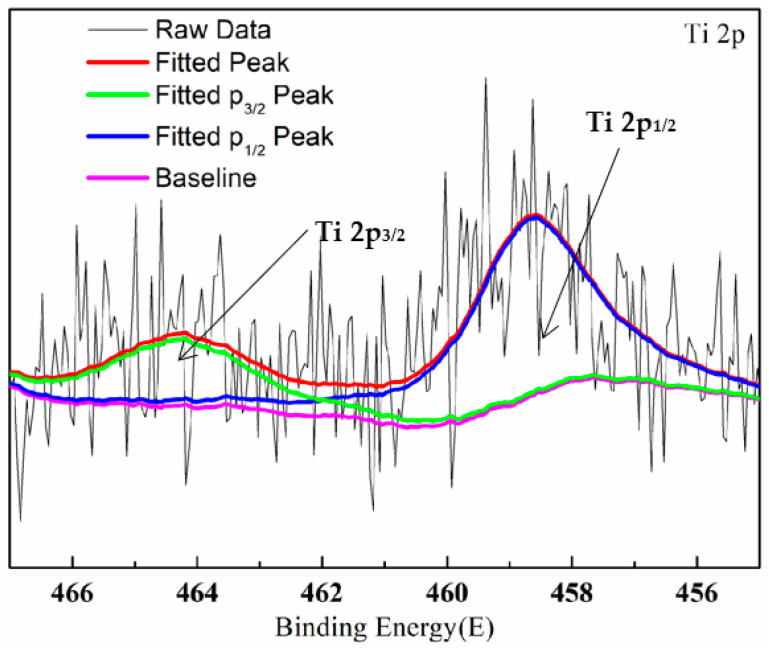
XPS diagram of Ti5-15 MAO film.

**Figure 8 materials-16-01830-f008:**
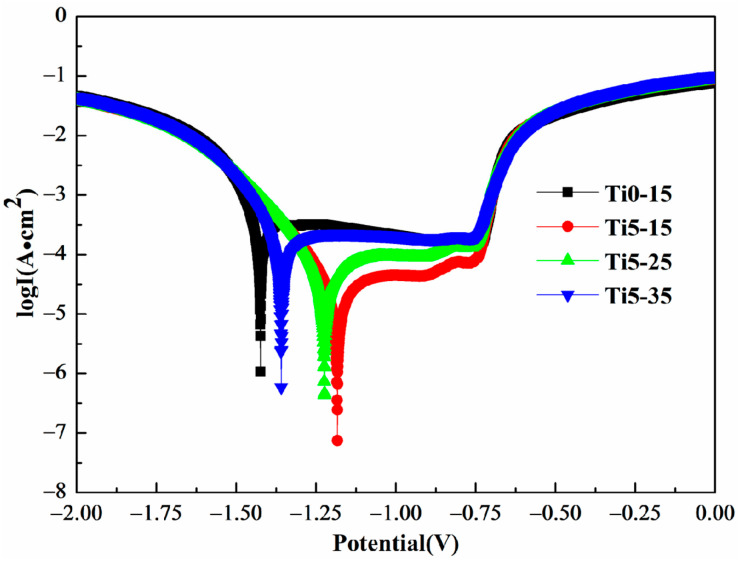
Potentiodynamic polarization curves of coatings Ti0-15, Ti5-15, Ti5-25 and Ti5-35.

**Figure 9 materials-16-01830-f009:**
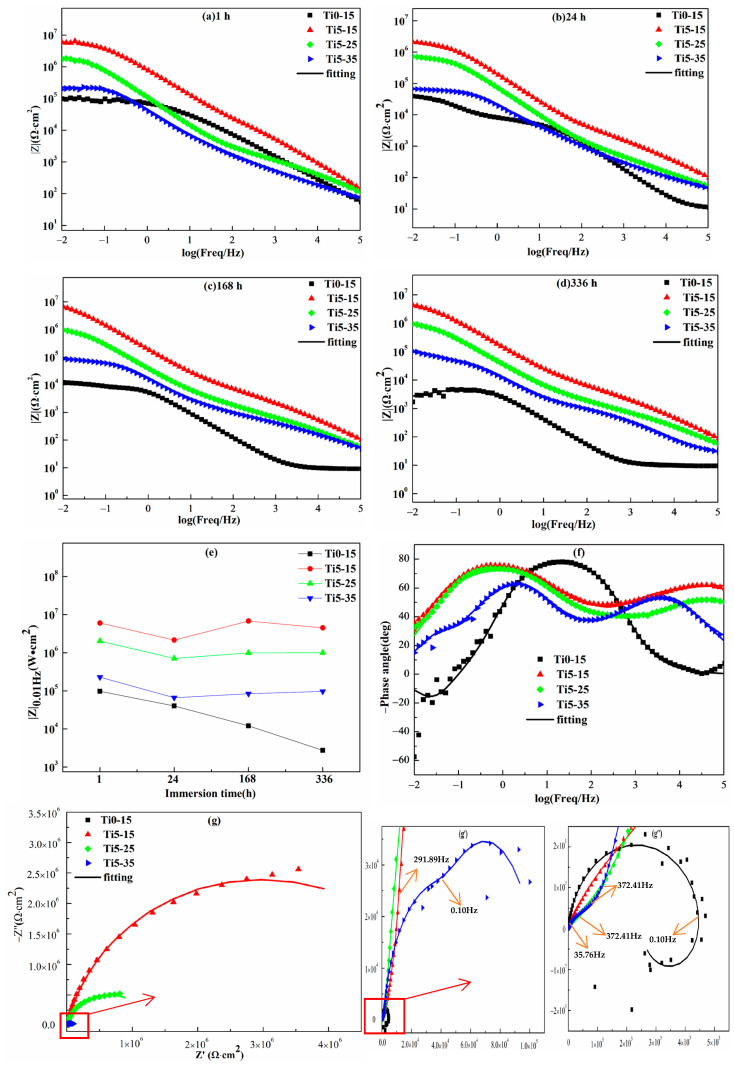
Bode impedance (**e**) of different MAO coatings treated in 3.5% sodium chloride corrosive medium after (**a**) 1 h, (**b**) 24 h, (**c**) 168 h and (**d**) 336 h. (**e**) |Z|0.01 Hz values of Ti0-15, Ti5-15, Ti5-25 and Ti5-35 after soaking. (**f**) Phase diagram, (**g**–**g**’’) Nyquist diagram of Ti0-15, Ti5-15, Ti5-25 and Ti5-35 after soaking for 336 h in 3.5% NaCl solution.

**Figure 10 materials-16-01830-f010:**
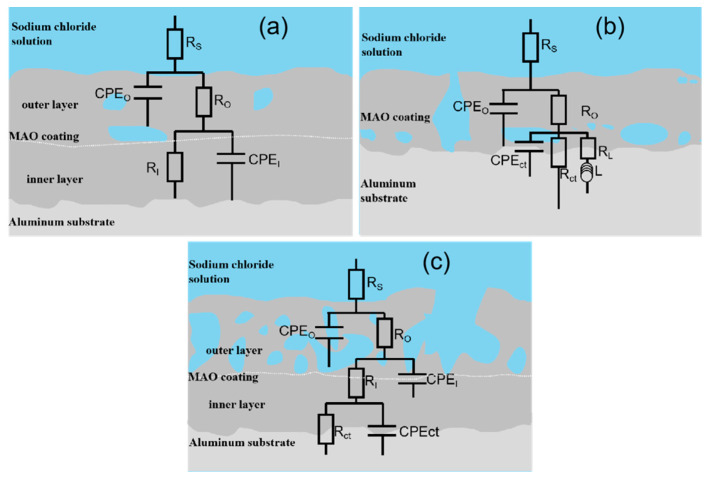
Equivalent circuits used to fit the EIS plots of (**a**) Ti5-15 and Ti5-25, (**b**) Ti0-15 and (**c**) Ti5-35.

**Figure 11 materials-16-01830-f011:**
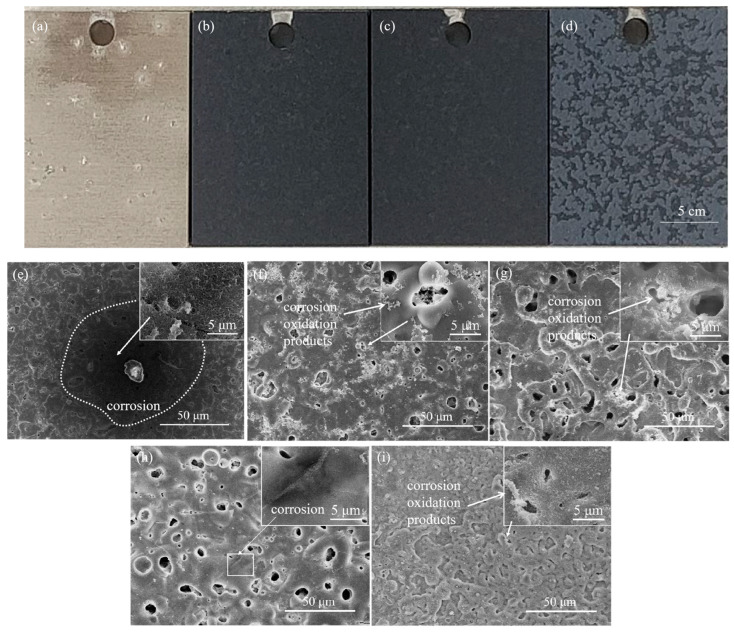
Digital photographs and SEM images of different samples after soaking in 3.5% sodium chloride solution for 336 h: (**a**,**e**) Ti0-15, (**b**,**f**) Ti5-15, (**c**,**g**) Ti5-25 and (**d**,**h**,**i**) Ti5-35.

**Figure 12 materials-16-01830-f012:**
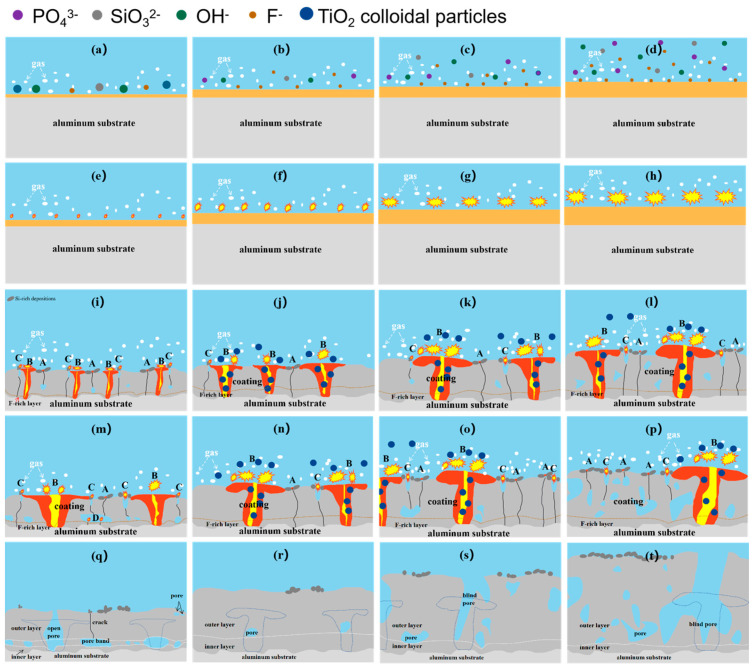
Growth mechanism model of the MAO coatings for different samples: (**a**,**e**,**i**,**m**,**q**) Ti0-15, (**b**,**f**,**j**,**n**,**r**) Ti5-15, (**c**,**g**,**k**,**o**,**s**) Ti5-25 and (**d**,**h**,**l**,**p**,**t**) Ti-35.

**Table 1 materials-16-01830-t001:** Identified codes of the MAO-treated samples and their corresponding electrolyte composition, temperature, conductivity and pH.

Sample/ Coating Code	Electrolyte Composition	Cooling Temperature	Conductivity (κ, mS ± 0.1)	pH
Ti0-15	10 g/L Na_2_SiO_3_, 10 g/L Na_3_PO_4_, 2 g/L NaF	15 °C	17.92	12.17
Ti5-15	10 g/L Na_2_SiO_3_, 10 g/L Na_3_PO_4_, 2 g/L NaF, 5 g/L K_2_TiF_6_	15 °C	16.79	8.18
Ti5-25	10 g/L Na_2_SiO_3_, 10 g/L Na_3_PO_4_, 2 g/L NaF, 5 g/L K_2_TiF_6_	25 °C	17.49	8.11
Ti5-35	10 g/L Na_2_SiO_3_, 10 g/L Na_3_PO_4_, 2 g/L NaF, 5 g/L K_2_TiF_6_	35 °C	17.65	8.03

**Table 2 materials-16-01830-t002:** EDS results of the surfaces of films Ti0-15, Ti5-15, Ti5-25 and Ti5-35.

Element/wt%	Al	O	Si	P	Na	F	Ti	K
Ti0-15	61.99	28.216	6.129	2.694	0.971	0	-	-
Ti5-15	35.733	29.581	4.523	4.181	1.245	0.516	23.535	0.686
Ti5-25	32.763	28.081	7.042	4.019	4.06	3.824	18.587	1.624
Ti5-35	26.54	27.728	8.904	4.982	8.17	4.086	17.034	2.557

**Table 3 materials-16-01830-t003:** EDS results of the coatings for samples with different treatment times.

Element/wt%	Al	O	Si	P	Na	F	Ti	K
90″-surface	58.073	23.413	1.926	2.198	0.148	9.383	4.606	0.242
90″-point1	51.224	31.886	3.147	2.695	0.358	4.965	5.255	0.166
90″-point2	69.767	16.374	0.936	1.014	0.124	10.194	1.536	0.055
130″-surface	39.577	30.77	4.723	3.162	0.622	1.032	19.924	0.19
130″-point3	41.043	26.731	3.694	3.112	0.611	3.896	20.855	0.058
130″-point4	34.810	37.370	6.841	3.075	0.668	0.278	16.814	0.124

**Table 4 materials-16-01830-t004:** Thickness of the MAO coating inner layer, outer layer and the whole coating of different samples after micro-arc oxidation for 10 min.

Coating	Inner Layer		Outer Layer		All Coating	
Sample Thickness (μm)	Average	Standard Deviation	Average	Standard Deviation	Average	Standard Deviation
Ti0-15	1.2	0.2	4.6	0.5	5.8	0.7
Ti5-15	3.9	0.3	8.5	0.8	12.3	1.0
Ti5-25	2.5	0.3	18.7	1.3	21.3	1.5
Ti5-35	2.0	0.2	44.0	4.0	47.9	4.1

**Table 5 materials-16-01830-t005:** Fitting data obtained from the polarization curves of coatings Ti0-15, Ti5-15, Ti5-25 and Ti5-35.

Sample	β_a_	β_c_	E_corr_	i_corr_	R_corr_
	(mV·dec^−1^)	(mV·dec^−1^)	(V)	(A·cm^−2^)	(mm·a^−1^)
Ti0-15	3096.2	147.95	−1.424	6.297 × 10^−4^	0.617
Ti5-15	11,609	223.62	−1.183	7.238 × 10^−5^	0.071
Ti5-25	3260.3	222.67	−1.224	1.354 × 10^−4^	0.133
Ti5-35	3930.2	173.9	−1.359	3.545 × 10^−4^	0.348

**Table 6 materials-16-01830-t006:** Electrochemical parameters of EIS of Ti0-15, Ti5-15, Ti5-25 and Ti5-35 soaked in 3.5% sodium chloride solution for 336 h.

Sample	R_S_	R_O_	R_I_	R_CT_	CPE_O_		CPE_I_		CPE_ct_		L	Chi Square
	(Ω·cm^2^)	(Ω·cm^2^)	(Ω·cm^2^)	(Ω·cm^2^)	Y_O_	n_O_	Y_I_	n_I_	Y_ct_	n_ct_	(H·cm^−2^)	
					(Ω^−1^·s^n^·cm^−2^)		(Ω^−1^·s^n^·cm^−2^)		(Ω^−1^·s^n^·cm^−2^)			
Ti0-15	10	120	-	226	1.61 × 10^−5^	0.92			3.71 × 10^−5^	0.91	950	9.74 × 10^−4^
Ti5-15	28	1.01 × 10^6^	5.94 × 10^6^	-	9.49 × 10^−7^	0.75	1.22 × 10^−6^	0.86	-	-	-	1.01 × 10^−3^
Ti5-25	47	1.3 × 10^5^	9.52 × 10^5^	-	6.26 × 10^−6^	0.74	4.37 × 10^−6^	0.87	-	-	-	1.51 × 10^−3^
Ti5-35	16	319	2196	7714	6.2 × 10^−6^	0.75	1.14 × 10^−5^	0.80	1.23 × 10^−6^	0.96	-	1.25 × 10^−3^

## Data Availability

The data presented in this study are available on request from the corresponding author. The data are not publicly available due to privacy.

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
