# Peer review of "Effects of K2TiF6 and Electrolyte Temperatures on Energy Consumption and Properties of MAO Coatings on 6063 Aluminum Alloy"

_materials, 2023, doi:10.3390/ma16051830_

Round 1

Reviewer 1 Report

Dear autors,

After carefully reading the scientific paper developed by you, I would like to address the following questions and observations:

1. The data from Table 1 – Why, if the 5g/L K2TiF6 compound were added, the effects will be a decrease in the conductivity of the electrolyte and its pH by 4 pH units (a pronounced decrease in the alkaline environment)?

2. Corrosion behavior of the MAO films was investigated only with the electrochemical impedance spectroscopy (EIS) method in 3.5 wt% NaCl? Why did you choose only this electrochemical method of investigation, knowing that it is not the most conclusive? I propose to repeat the corrosion studies using the linear polarization method (Tafel method) which will provide much more conclusive results for the study carried out. This method can provide direct information of both the corrosion current and a comparison of the samples obtained in the absence, respectively in the presence of the added K2TiF6 compound

3. “The corrosion tests used a platinum plate as the reference electrode and a saturated calomel electrode as the counter electrode and 1 cm2 of film that was exposed in NaCl solution as the working electrode.” Some informations presented in this paragraph are not correctly, I don't think platinum is the reference electrode but the counter electrode, and the calomel-saturated electrode certainly played the role of the reference electrode in these studies. Please check this information.

4. Why did you choose 336 hours (2 weeks) as immersion time in the corrosive environment for the studied samples?

5. ‘’Fig. 6e showed the variation of impedance values |Z|0.01Hz at 0.01 Hz with the prolonging of the immersion time.’’ What is the importance of this graph, respectively of the presented parameters?

Reviewer 2 Report

The authors reported the interesting results and conducted a significant work. The manuscript was well written and organized. However, there existed several issues that should be revised.

1. The section "Introduction" seems like an economic study, I think it could be improved.

2. Explain how you calculated the film thickness?

3. The choice of substrate and corrosive medium must be justified.

4. Why you did not use other more powerful techniques in characterization?

5. All parameters of electrochemical impedance spectroscopy should be discussed

6.  Comparison with previous works are not reported.

7. Please provide figures of high resolution.

Thus, the manuscript should experience the major revision before acceptance.

Reviewer 3 Report

Herein, the author has investigated a considerable decrease in the energy consumption of Micro-arc oxidation 11 (MAO) films on 6063 Al alloy achieved using policies based on K2TiF6 additive and electrolyte 12 temperatures control. I must say that the author has done appreciable and hard work to conduct this research but still, there is room for improvement, thus, I would suggest a minor revision of the manuscript to improve the quality of the article for its possible publication in this reputed journal. Kindly find my comments below:

1.     Highlights must be revised and relevant findings and observations should be included.

2.     The short mechanism mentioned in the abstract should be recast with the aid of insight findings.

3.     In the current state, the introduction lacks originality and must be expanded with the inclusion of recent literature and must state why there is a need to conduct this experiment and what is the contribution toward society or the industrial sector.

4.     Discuss more about micro arc oxidation with recent literature support in the introduction.

5.     Elaborate more for figure 3.

6.     Provide JCPDS data for figure 5.

7.     Fitting of figure 6 (g) required.

8.     References need to be revised and check. The author is suggested to add literature in the article: Coatings 12, no. 10 (2022): 1459, https://doi.org/10.3390/coatings12101459; Process Safety and Environmental Protection 161 (2022): 801-818, https://doi.org/10.1016/j.psep.2022.03.082

Round 2

Reviewer 2 Report

Reviewer # :  The authors reported the interesting results and conducted a significant work. The manuscript was well written and organized. However, there existed several issues that should be revised.

1.     Reformulate the abstract in order to clearly show the strengths of this work.

2.     The choice of substrate and corrosive medium must be justified.

3.     The experimental part must be detailed.

4.     Equation 1 needs reference.

5.     Nyquist plots should show some define frequencies.

6.     Why did the authors not measure XPS to strengthen the confidence of the results.

7.     Comparative analysis of the present data with those published in the literature for the similar type of compounds would support and can improve the quality of discussion

8.     What is the social contribution of your research? How your results and approaches are useful for industrial sectors?

Thus, the manuscript should experience the major revision before acceptance.
